# Exploring the Relationships Between Academic Engagement and Professional Suitability in Social Work Students: The Mediating Role of Academic Satisfaction

**DOI:** 10.3390/bs15111518

**Published:** 2025-11-08

**Authors:** José Luis Gálvez-Nieto, Guillermo Davinson-Pacheco, Julio Tereucán-Angulo, Ignacio Norambuena-Paredes, Claudio Briceño-Olivera, Ximena Briceño-Olivera, Vicenta Rodríguez-Martín

**Affiliations:** 1Departamento de Trabajo Social, Universidad de La Frontera, Temuco 4780000, Chile; jose.galvez@ufrontera.cl (J.L.G.-N.); guillermo.davinson@ufrontera.cl (G.D.-P.); julio.tereucan@ufrontera.cl (J.T.-A.); claudio.briceno@ufrontera.cl (C.B.-O.); ximena.briceno@ufrontera.cl (X.B.-O.); 2Programa de Doctorado en Ciencias Sociales, Universidad de La Frontera, Temuco 4780000, Chile; 3Facultad de Ciencias Sociales y Tecnologías de la Información, Universidad Castilla la Mancha, 28001 Madrid, Spain; vicenta.rodriguez@uclm.es

**Keywords:** academic engagement, ethical–practical suitability, academic satisfaction, university students, structural equation model

## Abstract

Understanding the factors that influence academic engagement and the perception of ethical–practical suitability is essential for improving university training processes in Social Work. In this context, academic satisfaction plays a key role. This study, with a cross-sectional design and a structural equation modelling (SEM) approach, aimed to examine the direct and indirect relationships among academic engagement, ethical–practical suitability, and academic satisfaction in a sample of Social Work students in Chile. A total of 298 Social Work students participated in this study, from 9 public and private universities (23.1% men, 76.9% women), with a mean age of 21.74 years (SD = 3.470). The results, obtained from a structural equation model, confirm that academic satisfaction significantly and partially mediates the relationship between ethical–practical suitability and academic engagement. Likewise, positive direct effects were observed among the three variables. Confirming a partial mediating effect of academic satisfaction in the relationship between professional suitability and academic engagement. The results are discussed in terms of their relevance for designing training strategies aimed at strengthening academic engagement and the perception of professional readiness in higher education.

## 1. Introduction

In the field of university education, strengthening academic engagement has been identified as a key factor in achieving successful academic trajectories and developing professional competencies ([65]; [83]; [93]). Particularly in disciplines such as Social Work, where preparation for ethical and reflective practice is essential, it becomes necessary to explore the elements that shape students’ perception of ethical–practical suitability ([62]; [63]; [113]).

In this regard, Social Work training programs face a series of challenges that directly impact the quality of the academic experience and the process of professional identity construction ([98]; [116]). The demands of the program, along with tensions arising from changing institutional contexts, the growing complexity of intervention settings, and early exposure to highly emotional situations, can significantly affect students’ motivation, well-being, and engagement ([96]; [107]; [111]).

In this context, ethical–practical suitability emerges as a central component in the comprehensive training of future social workers ([24]; [33]). This construct goes beyond the domain of technical knowledge, integrating essential dimensions such as ethical awareness, intrinsic motivation, sensitivity to social vulnerability, and the ability to exercise critical judgment in complex and dynamic scenarios ([23]; [56]; [120]).

From this perspective, Social Work education requires pedagogical processes that go beyond the mere acquisition of disciplinary knowledge, simultaneously promoting critical reflection, ethical commitment, and the student’s subjective engagement with the challenges inherent to professional practice ([1]; [4]; [95]). Consequently, students’ perception of ethical–practical suitability is closely linked to the construction of their professional identity, which is expressed through a sense of belonging to the discipline, the development of a self-image as a future professional, and the willingness to practice the profession in an ethical, reflective, and socially responsible manner ([3]; [57]; [109]).

In this regard, academic satisfaction has been recognized as a key mediating factor in university training processes ([35]; [90]), as it significantly influences motivation, subjective well-being, and student retention ([10]; [19]). In the case of Social Work education, a satisfactory academic experience not only enhances student engagement but also strengthens their confidence in their competencies, which is essential for addressing the ethical, social, cultural, and digital challenges of future professional practice ([56]; [64]; [77]).

In this context, academic satisfaction can be understood as a component that bridges academic engagement, conceived as the student’s active participation in their educational process, and the perception of ethical–practical suitability as an outcome of the educational journey ([29]; [97]). Therefore, a university environment that fosters meaningful experiences, creates a sense of achievement, and recognizes individual effort contributes to the consolidation of a more defined and confident vocational identity ([22]; [106]). In this way, academic satisfaction not only affects the student’s subjective well-being but also strengthens their academic engagement with a professional practice that is reflective, contextualized, and oriented toward social change ([27]; [53]; [75]).

Despite advances in understanding the factors that influence the development of successful academic trajectories ([32]; [68]), there remains a limited body of empirical research that integrally connects variables such as ethical–practical suitability for ethical practice, academic engagement, and academic satisfaction—particularly among Social Work students ([74]; [105]). This omission represents a critical gap, considering that these elements not only shape student well-being but also impact the quality of professional practice ([5]; [88]; [110]). By addressing these relationships in an integrated manner, it becomes possible to more precisely identify the factors that enhance comprehensive training and support the consolidation of a professional identity that is committed, reflective, and ethically grounded ([45]; [49]; [66]).

Although international literature has advanced understanding of academic engagement and academic satisfaction, a gap remains regarding how these processes are articulated with professional suitability in Latin American and Chilean contexts. In these educational environments, characterized by structural inequalities and tensions between vocation and employability, it is not yet clear how these variables interact to promote the well-being, motivation, and professional suitability of Social Work students ([30]; [84]). This lack of integrated evidence limits the understanding of the mechanisms that sustain ethically committed and socially situated formative trajectories.

In this regard, the present study offers a significant contribution to the field of Social Work by generating empirical evidence that helps to understand how the educational trajectories of those preparing to intervene in contexts of high social complexity are shaped ([17]; [44]; [67]). The findings may have direct implications for the design of pedagogical strategies, mentoring systems during professional practice processes, and student support mechanisms, fostering educational environments that are more responsive to students’ needs ([58]; [100]; [125]). In doing so, these strategies support not only students’ retention and well-being throughout their training process but also their preparation in critically and contextually grounded competencies for professional practice aimed at social change ([54]; [86]).

### 1.1. Relationship Between Ethical–Practical Suitability and Academic Satisfaction

Various studies have demonstrated a significant association between ethical–practical suitability and academic satisfaction in university students ([20]; [92]; [117]), showing that both constructs interact dynamically throughout the educational process ([112]). In this context, the perception of ethical–practical suitability promotes greater academic self-efficacy, sustained commitment to learning, and a positive evaluation of institutional support—factors that together strengthen student satisfaction with the university experience ([74]; [119]; [123]).

This relationship can be understood through the Person–Environment Fit Theory ([51]; [59]), which posits that the degree of congruence between the student’s individual characteristics—such as interests, skills, and goals—and the values and opportunities present in the educational environment directly influences their formative experience ([72]). From this perspective, when curricular content and learning experiences are perceived as consistent with future professional practice, students develop a sense of purpose and belonging that enhances their academic engagement ([11]; [108]; [114]). In this regard, ethical–practical suitability acts as a key facilitator of academic well-being by promoting adaptive coping styles, strengthening the perception of competence, and fostering more meaningful integration into the educational environment ([50]; [118]).

Therefore, both ethical–practical suitability and academic satisfaction are not only configured as relevant indicators of formative well-being ([61]) but also as facilitating factors for student engagement with their university journey and overall life satisfaction ([82]; [115]). A clearly defined professional identity strengthens the bond between the student and their discipline, promoting greater involvement in academic challenges, the development of transversal competencies, and long-term vocational projection ([69]; [72]). In turn, a satisfactory academic experience increases intrinsic motivation and reinforces the self-perception of suitability for future professional practice ([31]; [34]). Together, these elements shape a virtuous cycle that drives students to engage in their education actively, assuming their development as future professionals with greater responsibility, critical thinking, and ethical commitment.

### 1.2. Relationship Between Academic Engagement and Academic Satisfaction

Various studies have shown that academic engagement is a significant predictor of academic satisfaction, particularly among university students ([9]; [18]; [101]; [115]). This link is explained by the positive relationship between active involvement in academic tasks, institutional belonging, and the appreciation of the educational process ([19]; [36]; [35]; [91]). In this regard, high levels of academic engagement are consistently associated with a more satisfying and enriching educational experience ([12]; [21]; [60]).

In this way, the student’s active involvement in their formative processes becomes a decisive factor for the activation and consolidation of essential personal resources in university learning ([38]; [46]; [121]). Along these lines, academic engagement is positioned as a structuring component of educational development, as it facilitates not only the effective coping with academic demands but also the adoption of self-regulated strategies, the strengthening of meaningful support networks, and the progressive construction of an academic identity aligned with the demands of higher education ([8]; [90]).

From this perspective, the connection between academic engagement and academic satisfaction gains particular relevance in the university context ([40]; [94]; [101]). Firstly, academic engagement has been associated with higher levels of academic performance, persistence in university programs, and the development of transversal competencies such as self-regulation, autonomy, and resilience in the face of academic adversity ([16]; [70]).

Likewise, students with high levels of academic satisfaction tend to exhibit a greater sense of institutional belonging, lower intention to drop out, and a more positive attitude toward learning, which supports sustained academic paths aligned with their personal and professional goals ([15]; [26]; [97]). Together, these variables have an impact not only on academic performance but are also linked to psychological well-being, academic self-esteem, and the perception of self-efficacy, positioning them as key elements in promoting comprehensive academic success ([41]; [48]; [102]).

### 1.3. Relationship Between Academic Engagement and Ethical–Practical Suitability

Recent research has examined the interrelationship between academic engagement and ethical–practical suitability, as well as their combined influence on levels of academic satisfaction ([79]; [91]; [115]). The development of academic engagement in university students and professional suitability, specifically ethical–practical suitability, generates significant benefits in various areas ([25]; [52]; [109]). At the individual level, it enhances academic performance, increases satisfaction with the educational experience, and reduces student dropout rates ([47]; [102]). It also strengthens key skills such as planning, efficient time management, perseverance, and self-efficacy ([87]).

In parallel, academic engagement also promotes psychological well-being, as it is associated with lower levels of anxiety and depression ([55]; [85]). At the institutional level, academic engagement and ethical–practical suitability translate into improvements in key indicators such as student retention and timely graduation ([20]; [37]). Additionally, it fosters a favourable academic climate by creating collaborative and motivating learning environments ([124]).

Finally, from a social perspective, academic engagement and ethical–practical professional suitability foster attitudes and values in students that are later reflected in workplace contexts, such as responsibility, perseverance, and work ethics ([90]; [122]). Moreover, engaged students tend to develop a greater awareness of their social role, showing a higher predisposition toward civic and community participation ([73]).

In this regard, the interrelationship between academic engagement and ethical–practical ethical–practical suitability constitutes a fundamental axis for the comprehensive education of university students, as it links academic performance with the development of values, skills, and dispositions that are key not only for their personal well-being but also for their future employability and active participation in society ([2]).

Along these lines, recent research has shown that academic engagement not only represents a psychological commitment to educational goals but also acts as a significant mediator between academic resilience and academic performance ([78]). This means that students who persevere in the face of educational adversity and develop a sense of purpose and belonging toward their studies can translate their resilience into concrete and sustained achievements, while simultaneously strengthening their professional identity. Thus, fostering an academic culture that enhances resilience and ethical commitment to the learning process is essential for promoting successful and socially responsible educational pathways ([14]).

From a complementary theoretical perspective, it is pertinent to consider alternative frameworks that broaden the understanding of the proposed mediation model. In this regard, Self-Determination Theory ([99]) provides a solid foundation by emphasising that the satisfaction of psychological needs—such as autonomy, competence, and relatedness—enhances intrinsic motivation, sense of purpose, and sustained commitment to professional training. Complementarily, Person–Environment Fit Theory ([51]) highlights the importance of congruence between individual values, interests, and abilities and the conditions of the educational environment, shaping more coherent and satisfying academic experiences. Likewise, the Job Demands–Resources Model ([7]) helps explain the balance between academic demands and personal and institutional resources. This approach suggests that resources such as self-efficacy, teacher support, and intrinsic motivation foster well-being and engagement. Taken together, these approaches broaden the theoretical foundation of the model by positioning academic satisfaction as a mediating mechanism that translates the perception of professional suitability into higher levels of academic engagement and vocational sense.

### 1.4. Research Objective and Hypothesis

In relation to the background reviewed, this study aimed to address the gaps in university research on the factors that explain academic engagement, particularly in the context of Social Work education. Therefore, it proposes to examine the role that perceived ethical–practical suitability and academic satisfaction play in explaining academic engagement. While previous studies have documented associations between these constructs, there is still limited empirical evidence that simultaneously explores their direct and indirect relationships. For this reason, the general objective of this study is to examine the direct and indirect relationships between ethical–practical suitability, academic satisfaction, and academic engagement in a sample of Social Work students in Chile. Based on this objective, four hypotheses are proposed, as represented in Figure 1: (H1) ethical–practical suitability will be directly and positively associated with academic satisfaction; (H2) academic satisfaction will be directly and positively associated with academic engagement; (H3) ethical–practical suitability will be directly and positively associated with academic engagement; and (H4) academic satisfaction will mediate the relationship between ethical–practical suitability and academic engagement.

## 2. Materials and Methods

### 2.1. Participants

The study population consisted of 1306 undergraduate university students from 9 Social Work schools, both public and private, in Chile. The selection of participants was carried out through stratified probability sampling with variable sampling proportion, considering a 95% confidence level, a 5% standard error, and a variance of *p* = q = 0.5 ([103]). Each institution constituted a stratum, and the sample allocation was proportional to its enrollment size. Within each stratum, a simple random selection of students enrolled in the Social Work program was conducted.

Participation was voluntary and anonymous, coordinated with program directors and faculty members, who distributed the survey link hosted on the QuestionPro platform. The inclusion criteria were: being enrolled in the Social Work program, pursuing regular undergraduate studies, and providing informed consent to participate. No additional exclusion criteria were established.

The final sample included 1306 Social Work students from nine Chilean universities, both public and private. To ensure the integrity of the collected data, the mandatory response option was activated on the QuestionPro platform, preventing missing values. The sample consisted of students of both sexes (23.1% men and 76.9% women), aged 18–42 years (M = 21.74; SD = 3.47). Regarding ethnic self-identification, 21.4% reported belonging to an Indigenous group, while 78.6% did not. Regarding place of residence, 76% lived in urban areas and 24% in rural areas.

### 2.2. Instruments

First, a sociodemographic questionnaire was administered to gather identifying information about the students. The instrument included closed-ended questions regarding gender, age, ethnic group affiliation, university affiliation, and place of residence.

Additionally, the adapted version of the Professional Suitability Scale for Social Work Practice by [109] ([109]) for Chilean students was administered. This scale measures the extent to which Social Work students perceive themselves as prepared for professional practice. It consists of 18 items and three correlated factors: Ethical–Practice Suitability (12 items, e.g., I have believed in the value and dignity of each individual), Social Consciousness (3 items, e.g., I have demonstrated a commitment to social change), and Personal Suitability (3 items, e.g., I am able to manage negative life experiences). In this study, the Ethical–Practice Suitability factor was used, which showed an adequate level of reliability measured through McDonald’s omega coefficient (ω = 0.906) and an adequate validity indicator (Average Variance Extracted (AVE) = 0.537).

Additionally, a measure of academic engagement was administered, derived from the University Academic Performance Scale ([89]). The academic engagement measure assesses the degree of dedication to academic activities. It consists of five items (e.g., I dedicate time daily to complete the tasks assigned to me in my academic programme) and uses a 7-point response scale (1 = never, 7 = always). In this study, the instrument showed an adequate level of reliability measured through McDonald’s omega coefficient (ω = 0.826) and an adequate validity indicator (Average Variance Extracted (AVE) = 0.520).

Finally, the Spanish version of the Academic Satisfaction Scale ([42]) was administered. This scale measures the degree of academic satisfaction that students perceive. It is a self-report instrument composed of 8 items (e.g., I am interested in the classes, the professors are open to dialogue). This instrument was answered using a four-point ordinal scale (1 = never, 4 = always). In this study, the instrument showed an adequate level of reliability measured through McDonald’s omega coefficient (ω = 0.815) and an adequate validity indicator (Average Variance Extracted (AVE) = 0.543).

### 2.3. Procedure

For the administration of the instrument, contact was established with the authorities of the participating universities, who granted the corresponding institutional authorization. The study was approved by the Scientific Ethics Committee of the University of La Frontera (File UFRO No. 064_24). Data collection was conducted between September and December 2024, using an online questionnaire hosted on the QuestionPro platform.

The research team distributed the invitation to participate via email, which included the informed consent form and a direct link to the questionnaire. This document specified the objectives of the study, the voluntary nature of participation, guarantees of confidentiality and anonymity, the absence of associated risks, and the participants’ right to withdraw from the study at any time without consequences.

Additionally, three expert judges from Chile evaluated the items corresponding to the Suitability, Academic Satisfaction, and Academic Engagement scales, with the aim of assessing their semantic and cultural appropriateness. After a thorough review process, the judges concluded that the items were understandable and relevant to the context of the application.

### 2.4. Data Analysis

To achieve the objectives of this study, descriptive and correlational analyses were first conducted using SPSS v.25 software. Subsequently, structural equation models (SEM) were estimated using Mplus v.8.11 software ([76]). For parameter estimation, the Weighted Least Squares Mean and Variance adjusted method (WLSMV) was used, which is recognized for its suitability when working with ordinal variables ([6]).

The structural equation model included three latent factors: professional suitability, academic satisfaction, and academic engagement. Direct relationships between these factors were specified, and the presence of a mediation effect was evaluated using the MODEL INDIRECT command in Mplus. This command allows for the simultaneous calculation of the model’s total, direct, and indirect effects, incorporating statistical significance tests (*p*-value) and 95% confidence intervals.

The evaluation of model fit was based on various goodness-of-fit indices, including the Comparative Fit Index (CFI) and the Tucker–Lewis Index (TLI), with values equal to or greater than 0.90 considered indicative of adequate fit ([104]). Additionally, the Root Mean Square Error of Approximation (RMSEA) and the Standardized Root Mean Square Residual (SRMR) were analyzed, with values equal to or less than 0.08 considered acceptable according to established criteria ([43]; [28]).

Regarding the interpretation of structural effects, the cutoff points proposed by [71] ([71]) were adopted, classifying the effects as low (<0.30), moderate (between 0.30 and 0.50), and high (>0.50).

## 3. Results

### 3.1. Preliminary Analysis

Table 1 presents the Pearson r correlations among the study factors and the descriptive statistics for each variable. The results show positive and statistically significant associations between the factors. Specifically, Ethical-Practice Suitability showed a low-magnitude correlation with Academic Satisfaction (r = 0.275, *p* < 0.001) and a moderate-magnitude correlation with Academic Engagement (r = 0.321, *p* < 0.001). In turn, Academic Satisfaction presented a moderate association with Academic Engagement (r = 0.475, *p* < 0.001). These results indicate that higher perceptions of ethical-practice suitability are meaningfully related to greater academic satisfaction and engagement, with effects ranging from small to moderate according to conventional benchmarks. At the bottom of Table 1, the descriptive statistics for each factor are provided, allowing for a more detailed characterization of their behaviour in the sample.

### 3.2. Evaluation of the Structural Mediation Model

Figure 2 shows the results of the structural model with standardized coefficients. According to the evaluated indices, the model demonstrated satisfactory fit (WLSMV-χ^2^ (df = 272) = 567.955, *p* < 0.001; CFI = 0.949, TLI = 0.943, SRMR = 0.064; RMSEA = 0.065 [90% CI = 0.058–0.073]).

Regarding the structural relationships, the results confirmed the four proposed hypotheses. Ethical-practice suitability had a moderate, positive, and statistically significant effect (β_1_ = 0.379, *p* < 0.001) on academic satisfaction, indicating that higher perceptions of ethical–practical suitability are associated with greater levels of academic satisfaction. In turn, academic satisfaction exhibited a strong, positive effect (β_2_ = 0.495, *p* < 0.001) on academic engagement, while suitability also showed a small-to-moderate direct effect (β_3_ = 0.265, *p* < 0.001) on engagement.

Additionally, the indirect effect from suitability to academic engagement through academic satisfaction was moderate and statistically significant (β_4_ = 0.188, 95% CI [0.124–0.258]), representing 42% of the total effect and indicating partial mediation. This highlights the mediating role of academic satisfaction in the relationship between perceived suitability and active engagement in academic activities.

Finally, the total effect of suitability on academic engagement was moderate and significant (β_total = 0.452, *p* < 0.001), resulting from the sum of paths β_3_ and β_4_. This finding underscores its relevance as a key explanatory factor of academic engagement, both directly and indirectly.

To examine the robustness of this mediation pattern, we compared the partial mediation model with two alternative specifications: a full mediation model (without the direct path from suitability to engagement) and a non-mediation model (only the direct path). The partial mediation model showed the best fit (CFI = 0.949; RMSEA = 0.065), significantly outperforming both the full mediation model (ΔCFI = 0.019; DIFFTEST *p* < 0.001) and the non-mediation model (ΔCFI = 0.054; DIFFTEST *p* < 0.001). These results provide strong empirical support for the partial mediation structure, indicating that both direct and indirect paths contribute significantly to academic engagement (Table 2).

## 4. Discussion

The results of the present study make it possible to affirm that ethical–practical suitability constitutes a central component in the formative experience of students, with significant effects both on their academic satisfaction and on their level of academic commitment. This empirical evidence rethinks a reductionist view of professional training, which is limited to the development of technical skills, and recognizes the importance of building educational environments in which students perceive coherence, relevance, and meaning in their academic trajectory. As authors such as [24] ([24]) and [3] ([3]) have pointed out, ethical, contextualized, and socially committed training not only strengthens professional identity but also increases students’ willingness to become actively involved in their educational process.

This study aimed to examine the direct and indirect relationships between academic engagement, ethical–practical suitability, and academic satisfaction in a sample of Social Work students from Chile. The results confirm the initial hypotheses, highlighting the need to promote these competencies to benefit the university environment and the curriculum. It is worth noting that regarding the relationship between the perception of ethical–practical suitability and academic satisfaction, the results confirm that the recognition of training consistent with values such as justice, responsibility, and social commitment is positively associated with a more satisfactory academic experience.

These findings are consistent with previous research that highlights the role of teacher credibility and formative legitimacy in the perception of satisfaction ([10]; [36]). As stated by [15] ([15]), academic satisfaction is shaped not only by curricular factors but also by an experience of ethical and emotional recognition of the training received. This study, focused on Social Work students in Chile, reaffirms that the perception of ethical–practical suitability is not a peripheral element, but a key determinant of academic satisfaction.

Regarding the link between academic satisfaction and academic engagement, the findings show a direct and significant relationship. Beyond conceiving satisfaction as a transitory emotional state, its mediating role in strengthening academic involvement is recognized. This interpretation coincides with what was proposed by [9] ([9]), who emphasizes that satisfaction is connected with processes of meaning, belonging, and intrinsic motivation, which are essential to sustain academic effort over time. Likewise, studies such as those by [16] ([16]) and [27] ([27]) have shown that subjective well-being and a positive evaluation of the educational experience are robust predictors of sustained engagement.

Regarding the relationship between ethical–practical suitability and academic engagement, the results of the structural model show that this association is significant. This relationship suggests that when students perceive that the training they receive is aligned with substantive professional values, a commitment is activated that transcends mere academic obligation. The literature on professional training in Social Work has highlighted that the congruence between the values of the academic program and the student’s expectations reinforces the sense of belonging and the construction of a committed professional identity ([13]; [116]; [74]). In this context, academic engagement is not limited to fulfilling institutional tasks or goals but is manifested as a form of value-based and reflective adherence to the educational project.

Finally, the analyses of the structural model confirm the mediating role of academic satisfaction between ethical–practical suitability and academic engagement. This mediation reveals a psychological mechanism, where the impact of training perceived as adequate in ethical and practical terms is channelled through a subjectively meaningful experience. As recent studies suggest ([18]; [90]; [35]), satisfaction not only generates well-being but also enhances involvement by activating motivational, emotional, and contextual appreciation processes within the educational setting ([39]). Consequently, strengthening student engagement requires not only ensuring technical quality standards but also cultivating formative experiences that students perceive as fair, rewarding, and consistent with their professional aspirations ([80]).

Beyond empirical confirmation of the proposed relationships, it is necessary to deepen the interpretation of the role of ethical–practical suitability in students’ formative experience. From a theoretical-disciplinary perspective, this construct can be understood as an articulating axis between the internalization of vocational values and the progressive configuration of professional identity ([109]). In the field of Social Work, the perception of coherence among curricular content, pedagogical practices, and the ethical principles that guide the profession fosters a sense of belonging and purpose that transcends mere technical competency acquisition ([17]). In this sense, ethical–practical suitability operates as a formative device that strengthens students’ intrinsic motivation and vocational commitment, shaping a type of academic engagement grounded in reflection and in identification with the emancipatory aims of the discipline ([74]; [81]).

In summary, these findings provide empirical evidence that contributes to a broader and deeper understanding of academic engagement by positioning the ethical dimension of training as a unifying axis between satisfaction and ethical–practical suitability. From this perspective, designing training programs that integrate ethical education transversally, not as an isolated component but as a structural orientation of the curriculum, teaching practices, and participatory spaces, represents a key strategy to promote more meaningful, fair, and sustainable academic trajectories ([2]; [23]; [49]).

### Limitations and Future Research

Despite the significant findings and the theoretical and practical implications of the study, it is necessary to acknowledge certain limitations. First, the cross-sectional design prevents establishing causal relationships between the variables studied, as the observed associations could be influenced by contextual or individual factors ([12]). In this regard, longitudinal studies would allow for a more accurate assessment of the directionality and stability of the effects observed over time.

Second, the sample consisted exclusively of Social Work students from a Chilean university, which limits the generalization of the findings to other majors, disciplines, or cultural contexts ([24]). Perceptions of ethical–practical suitability, academic satisfaction, and academic engagement may vary substantially across professional fields with different normative and pedagogical orientations. It is also important to acknowledge the potential self-report bias and common method variance inherent in single-source survey designs, which may have minor implications for interpreting the results. Finally, the gender imbalance observed in the sample, characteristic of Social Work education, may have influenced certain response patterns, given that formative experiences and perceptions of professional suitability can differ by gender. In this regard, it is recommended that future research consider more balanced sampling strategies that explore possible gender differences and nuances in the construction of professional suitability and academic well-being.

In line with the above, future research could address these limitations through longitudinal designs that allow for examining how the perception of ethical–practical suitability evolves throughout academic training, and how it has a sustained impact on academic satisfaction and academic engagement. Likewise, it would be relevant to replicate the model in more diverse samples, incorporating students from different areas of knowledge and from international educational contexts, which would make it possible to verify the external and cultural validity of the proposed model ([23]).

From the international literature, [109] ([109]) highlight that the relevance of professional suitability transcends individual evaluation, as it is directly linked to curricular development and institutional practice in Social Work education ([25]). The results derived from the assessment of this competence provide feedback to academic programs, guiding adjustments in content, methodologies, and practice experiences to strengthen coherence between professional values and the demands of real practice ([61]; [66]).

In this regard, the implications for curriculum design in Social Work education are especially relevant. Deepening this dimension by incorporating concrete pedagogical recommendations would strengthen the study’s applied value and its usefulness for university educational practices. In this sense, the development of academic and ethical tutoring programs, reflective and experiential learning modules, and faculty mentoring spaces oriented toward the construction of a critical and committed suitability is suggested. These strategies would allow for the integration of ethical–practical suitability as a transversal axis of the curriculum, fostering formative processes that articulate technical knowledge, ethical reflection, and social commitment from the early years of professional training.

In this way, the transversal incorporation of ethical–practical suitability into Social Work training processes constitutes a strategic path to strengthen coherence between ethical principles, critical reflection, and professional practice. This approach makes it possible to consolidate comprehensive training that integrates knowing, doing, and being, promoting in future professionals an ethical and transformative commitment to the social realities in which they intervene.

Finally, it is suggested to delve deeper into the exploration of moderating and mediating factors that influence these relationships ([14]; [20]). Variables such as teacher support, institutional sense of belonging, academic self-efficacy, or academic resilience ([9]) could offer a more complex understanding of the process through which ethical training impacts the student experience. This type of analysis will make it possible to guide pedagogical and curricular interventions that strengthen the link between professional training and academic well-being.

## 5. Conclusions

The results of this study provide relevant empirical evidence regarding the role of ethical–practical suitability in the training of Social Work students, as it significantly influences both their academic satisfaction and their commitment to the educational process. In a professional context oriented toward social change and justice, these results affirm the need for training programs not only to transmit theoretical and technical knowledge but also to cultivate an ethical sense and professional belonging from the early years of training. The identification of a mediating effect of academic satisfaction between ethical–practical suitability and academic engagement allows for an understanding of how coherent and meaningful formative experiences strengthen the involvement of future professionals, fostering more consistent and vocationally integrated educational trajectories.

From a disciplinary perspective, this study constitutes a substantial contribution to the field of social work by making visible that ethical training should not be conceived as a complementary component, but as a structuring axis of the curriculum and pedagogical relationships. Incorporating ethical–practical suitability transversally into teaching–learning processes fosters the construction of a professional identity committed to the principles that govern the practice of Social Work. Consequently, these results provide an empirical basis for guiding curricular policies, institutional strategies, and teaching practices that strengthen formative coherence, student well-being, and the development of socially responsible future professionals.

In practical terms, these findings underscore the importance of Social Work universities and educators intentionally designing pedagogical environments that integrate ethical reflection with experiential learning. Strengthening tutoring systems, reflective supervision, and practice-based learning opportunities can enhance students’ perception of ethical–practical suitability, thereby reinforcing their academic satisfaction and formative commitment. By promoting these conditions, institutions will contribute to the training of socially conscious and ethically committed professionals, capable of responding to contemporary social challenges.

## Figures and Tables

**Figure 1 behavsci-15-01518-f001:**
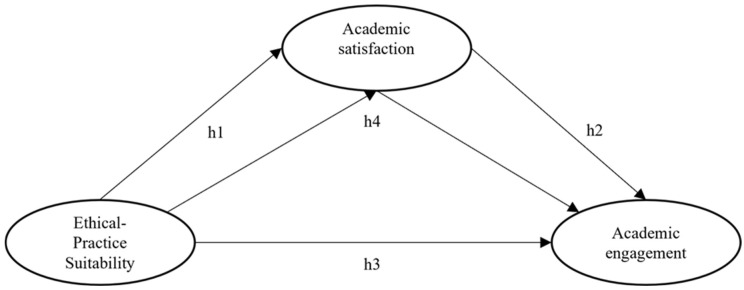
Theoretical framework of the relationships between variables.

**Figure 2 behavsci-15-01518-f002:**
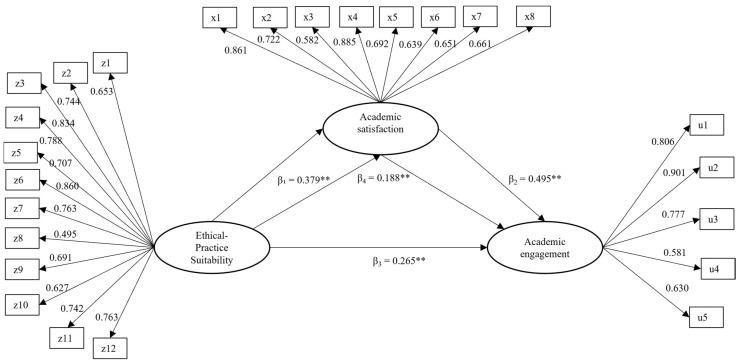
Structural model with standardized coefficients. Note: Rectangles represent observed variables (items), and ellipses represent latent constructs. Standardized factor loadings and structural path coefficients are shown. All estimates are statistically significant at ** *p* < 0.001.

**Table 1 behavsci-15-01518-t001:** Pearson’s r correlation matrix and descriptive statistics.

Factors	Ethical-Practice Suitability	Academic Satisfaction	Academic Engagement
Ethical-Practice Suitability	1		
Academic Satisfaction	0.275 **	1	
Academic engagement	0.321 **	0.475 **	1
**Descriptive Statistics**	**Ethical-Practice Suitability**	**Academic Satisfaction**	**Academic Engagement**
Media	51.53	25.07	25.55
Sd	6.47	3.64	5.38
g1	−2.17	−0.47	−0.20
g2	10.28	0.85	−0.72

Note: ** = *p* < 0.001; Sd, standard deviation; g1, skewness; g2, kurtosis.

**Table 2 behavsci-15-01518-t002:** Comparison of structural mediation models.

Model	χ^2^ (gL)	CFI	TLI	RMSEA [90% IC]	SRMR	ΔCFI vs. Parcial	DIFFTEST *p*	Interpretation
Partial mediation	567.955 (272)	0.949	0.943	0.065 [0.058–0.073]	0.064	—	—	Best overall fit
Full mediation	677.627 (273)	0.930	0.923	0.076 [0.069–0.084]	0.073	0.019	<0.001	Significantly poorer fit
No mediation	877.337 (273)	0.895	0.885	0.093 [0.086–0.100]	0.088	0.054	<0.001	Worst fit, lower explanatory parsimony

Note. CFI = Comparative Fit Index (CFI); TLI = Tucker–Lewis Index; RMSEA = Root Mean Square Error of Approximation; SRMR = Standardized Root Mean Square Residual. The chi-square difference tests were performed using DIFFTEST with the WLSMV estimator. ΔCFI ≥ 0.01 was considered evidence of a significant difference in model fit.

## Data Availability

The data presented in this study are available on request from the corresponding author due to ethical restrictions.

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
