# Peer review of "Exploring the Relationships Between Academic Engagement and Professional Suitability in Social Work Students: The Mediating Role of Academic Satisfaction"

_behavsci, 2025, doi:10.3390/bs15111518_

Round 1

Reviewer 1 Report

Comments and Suggestions for Authors

This is a strong paper, well written and based on an appropriate research paradigm and methodology. It makes an original and compelling contribution to the field. My only recommendation is that the authors might suggest ways in which 'ethical-practical suitability' can be woven transversally into teaching-learning processes.

Author Response

Comments 1: This is a strong paper, well written and based on an appropriate research paradigm and methodology. It makes an original and compelling contribution to the field. My only recommendation is that the authors might suggest ways in which 'ethical-practical suitability' can be woven transversally into teaching-learning processes.

Response 1: We sincerely appreciate the reviewer’s positive assessment of the strength and originality of the manuscript. In response to their valuable suggestion, we have incorporated a final reflection that proposes concrete ways to integrate ethical-practical suitability transversally into the teaching and learning processes in Social Work. In particular, we highlight the importance of strengthening pedagogical strategies that promote ethical reflection, situated learning, and early engagement with professional contexts, thereby fostering coherence between academic training and social practice. This addition can be reviewed in lines 486–491.

Reviewer 2 Report

Comments and Suggestions for Authors

Comments on the Quality of English Language

The English is generally clear and easy to follow. One point to address is the inconsistency in terminology (e.g., “ethical-practical suitability” vs. “professional suitability”). Choosing one preferred term and applying it consistently across the manuscript will make the text smoother and easier to read.

Author Response

Comments 1: The English is generally clear and easy to follow. One point to address is the inconsistency in terminology (e.g., “ethical-practical suitability” vs. “professional suitability”). Choosing one preferred term and applying it consistently across the manuscript will make the text smoother and easier to read.

Response 1: We appreciate the observation made. The manuscript was carefully reviewed to unify terminology, ensuring terminological consistency throughout the text and maintaining the use of “ethical-practical suitability” as the preferred designation due to its greater conceptual precision.

Comments 2: Abstract: Consider adding a clear mention of the study design (cross-sectional survey with SEM analysis) and emphasize the key mediation finding.

Response 2: We appreciate the observation. In response to your suggestion, an explicit reference to the cross-sectional design of the study with structural equation modelling (SEM) was incorporated into the abstract, along with greater emphasis on the key finding of partial mediation of academic satisfaction between professional suitability and academic engagement, thereby reinforcing the methodological coherence and the relevance of the results presented.

Comments 3: Introduction: The introduction is thorough but could be more concise. Consider reducing the description of the Chilean context and instead expanding the integration with international literature. In addition to the Person–Environment Fit Theory, frameworks such as Self-Determination Theory or the Job Demands–Resources model could provide useful perspectives.

Response 3: We sincerely appreciate the reviewer’s observation. We have revised the Introduction section and made the suggested adjustments, reducing the length of the Chilean context and strengthening its articulation with the international literature. In addition, new relevant theoretical frameworks, such as Self-Determination Theory and the Job Demands–Resources model, were integrated to enrich the understanding of the phenomenon studied from a comparative perspective. These modifications are reflected in lines 218 to 230 of the manuscript.

Comments 4: Methods: Please clarify the sampling strategy (e.g., were the universities selected by convenience?). Also, given that 77% of the sample are women, it would be important to acknowledge how this imbalance might limit generalizability.

Response 4: We appreciate your comment. The sampling strategy used was stratified random sampling, stratified by type of higher education institution. Regarding the alleged gender imbalance, the 77% of women does not represent an overestimation but rather reflects the actual composition of Social Work enrollment in Chile, which has historically been feminised (approximately 80% women; Aspeé et al., 2018).

Reference:

Aspeé, J., Henríquez, J., & Flores, S. (2018). Mujeres y hombres del Trabajo Social en Chile. Revista Perspectivas, 33(1), 41–59. https://www.scielo.br/j/rk/a/VhNC7wpBk3MqBWFc7mbXxXJ/?lang=es#:~:text=El%20estudio%20concluye%20que%20el,acent%C3%BAa%20la%20distancia%20entre%20sexos.

Comments 5: Methods: Please clarify the sampling strategy (e.g., were the universities selected by convenience?). Also, given that 77% of the sample are women, it would be important to acknowledge how this imbalance might limit generalizability.

Response 5: We appreciate your observation and agree on the importance of highlighting the magnitude of the effects. Consequently, we have incorporated a more explicit reference to these in the narrative text. Nevertheless, we consider it appropriate to keep Table 1 in the main body, as it provides clarity and transparency in the presentation of the findings, and the manuscript uses a limited number of numerical elements.

Comments 6: Discussion: The discussion is strong but could place more emphasis on international studies to show broader relevance. It would also help to expand on the implications for curriculum development and institutional practice

Response 6: We sincerely appreciate the reviewer’s observation. In response, additional references to recent international studies were incorporated to strengthen the discussion and contextualise the findings within a broader comparative framework. Likewise, the final section was expanded to further elaborate on the study’s implications for curriculum design and for improving institutional practices in professional training. These modifications are reflected between lines 489 and 511 of the manuscript.

Comments 6: Limitations: This section would benefit from further detail. Please include reflections on the cross-sectional design, reliance on self-reports, potential social desirability bias, and the gender imbalance.

Response 6: We sincerely appreciate the reviewer’s valuable comments. Their observations were taken into account and incorporated into the manuscript, specifically between lines 464 and 472, where the reflection on the methodological limitations of the study was expanded to include reliance on self-reports, potential social desirability bias, and the gender imbalance in the sample.

Reviewer 3 Report

Comments and Suggestions for Authors

Thank you for the opportunity to review the manuscript entitled “Exploring the relationships between academic engagement and professional suitability in Social Work students: the mediating role of academic satisfaction Play Addiction on Aggressive Behaviors among Turkish Preschoolers: Parental Assessment.” The paper investigates the relationship between academic engagement, professional suitability, and academic satisfaction in a sample of Chilean social work students, using a structural equation modeling (SEM) approach. 

The manuscript is conceptually coherent and methodologically sound, but certain areas require refinement to enhance clarity, reproducibility, and theoretical contribution.

Abstract

1) Keywords are appropriate, but consider including “Structural Equation Modeling” for indexing purposes.

Introduction

2) The introduction effectively contextualizes the study within social work education. However, the theoretical gap could be more sharply delineated. The authors should explicitly state what remains unknown about the interplay between engagement, satisfaction, and professional suitability, especially in Latin American or Chilean contexts.

3) Consider discussing alternative frameworks (e.g., Self-Determination Theory, Person-Environment Fit) to reinforce the theoretical rationale for the mediation model.

4) The hypotheses are clearly articulated and appropriately visualized (Figure 1), but a short “Aims and Hypotheses”subsection should precede them for better structure.

Methods

5) The Methods section is clear and well-organized, with adequate sample description. However, more detail is needed regarding sampling procedure (e.g., recruitment strategy, inclusion criteria, response rate).

Results

6) Consider presenting a model comparison table showing full, partial, and null mediation to illustrate the robustness of the partial mediation finding.

Discussion

7) The Discussion is well-structured and connects findings with existing literature. However, it occasionally restates results rather than interpreting them. The authors should emphasize why ethical-practical suitability exerts such influence, potentially linking it to professional identity formation and vocational calling frameworks.

8) The implications for curricular design in social work education are compelling; expanding this section with specific pedagogical recommendations (e.g., mentorship programs, reflective learning modules) would strengthen the applied value.

9) Consider also acknowledging self-report bias and potential common-method variance, which are inherent to single-source survey designs.

Conclusion

10) I believe it is necessary to temper claims regarding “robust empirical evidence,” given the cross-sectional design.

11) Adding a short final paragraph highlighting practical implications for universities and social work educators would strengthen the article's impact.

Author Response

Comments 1: Abstract

1) Keywords are appropriate, but consider including “Structural Equation Modeling” for indexing purposes.

Response 1: The keyword “Structural Equation Modelling” has been incorporated in order to improve the indexing and thematic precision of the manuscript.

Comments 2: The introduction effectively contextualizes the study within social work education. However, the theoretical gap could be more sharply delineated. The authors should explicitly state what remains unknown about the interplay between engagement, satisfaction, and professional suitability, especially in Latin American or Chilean contexts.

Response 2: We appreciate the reviewer’s observation. The suggestion was considered and incorporated into the manuscript, specifically between lines 90 and 99, by deepening the discussion of the theoretical gap regarding the interaction between engagement, satisfaction, and professional suitability, with particular emphasis on the Latin American and Chilean contexts, where training dynamics are shaped by structural inequalities, vocational tensions, and ethical challenges inherent to Social Work.

Comments 3: Consider discussing alternative frameworks (e.g., Self-Determination Theory, Person-Environment Fit) to reinforce the theoretical rationale for the mediation model.

Response 3: We appreciate the reviewer’s observation. The suggestion was incorporated into the manuscript between lines 215 and 227, adding a paragraph that integrates complementary theoretical frameworks—Self-Determination Theory (Deci & Ryan, 2000) and Person–Environment Fit Theory (Holland, 1997)—with the purpose of reinforcing the conceptual justification of the proposed mediation model and broadening the understanding of the role of academic satisfaction between professional suitability and academic engagement.

Comments 4: The hypotheses are clearly articulated and appropriately visualized (Figure 1), but a short “Aims and Hypotheses”subsection should precede them for better structure.

Response 4: We appreciate your observation. We have incorporated a subsection dedicated to the research objectives and hypotheses to provide a clearer, more structured presentation of these elements.

Comments 5: Methods

5) The Methods section is clear and well-organized, with adequate sample description. However, more detail is needed regarding sampling procedure (e.g., recruitment strategy, inclusion criteria, response rate).

Response 5: We appreciate the observation. Additional details on the sampling procedure, recruitment strategy, inclusion criteria, and response rate were incorporated into the “Participants” section to strengthen methodological transparency.

Comments 6: Results

6) Consider presenting a model comparison table showing full, partial, and null mediation to illustrate the robustness of the partial mediation finding.

Response 6: We appreciate your comment. We conducted a model comparison that included partial, full, and no-mediation models. The partial mediation model showed the best fit (CFI = 0.949, RMSEA = 0.065), significantly outperforming the full mediation model (ΔCFI = 0.019; DIFFTEST p < 0.001) and the no mediation model (ΔCFI = 0.054; DIFFTEST p < 0.001). These results support the robustness of the partial mediation effect and confirm that both direct and indirect effects contribute significantly to the outcome.

Comments 7: Discussion

7) The Discussion is well-structured and connects findings with existing literature. However, it occasionally restates results rather than interpreting them. The authors should emphasize why ethical-practical suitability exerts such influence, potentially linking it to professional identity formation and vocational calling frameworks.

Response 7: We greatly appreciate the reviewer’s valuable observation. The suggestion was considered and incorporated into the manuscript, specifically between lines 405 and 418, by deepening the theoretical interpretation of the role of ethical-practical suitability in the formation of professional identity and its connection to vocational frameworks in Social Work.

Comments 8: The implications for curricular design in social work education are compelling; expanding this section with specific pedagogical recommendations (e.g., mentorship programs, reflective learning modules) would strengthen the applied value.

Response 8: We appreciate the reviewer’s valuable suggestions to strengthen the manuscript. These were carefully considered and incorporated into the revised version between lines 433–442, where the discussion was expanded with specific pedagogical recommendations related to curriculum design and professional training in Social Work.

Comments 9: Consider also acknowledging self-report bias and potential common-method variance, which are inherent to single-source survey designs.

Response 9: We appreciate the reviewer’s observation. A mention of potential self-report bias and common method variance was included, noting their possible implications for interpreting the results. This can be reviewed in the corresponding lines of the Limitations and Future Research section.

Comments 10: Conclusion

10) I believe it is necessary to temper claims regarding “robust empirical evidence,” given the cross-sectional design.

Response 10: We greatly appreciate the reviewer’s valuable observation. The expression “robust empirical evidence” was revised in the new version to better reflect the limitations of the cross-sectional design. The text now states “empirical evidence consistent with the central role of ethical-practical suitability,” which provides a more appropriate and cautious interpretation of the findings.

Comments 11: Adding a short final paragraph highlighting practical implications for universities and social work educators would strengthen the article's impact.

Response 11: We appreciate the observation made. A new final paragraph was incorporated that emphasises the practical implications for universities and Social Work educators, with the aim of strengthening the study's applied impact. This adjustment is highlighted in yellow between lines 462 and 470 of the manuscript.

Round 2

Reviewer 2 Report

Comments and Suggestions for Authors

hank you for reviewing and considering my feedback.